# Structure and Formation of Z-DNA and Z-RNA

**DOI:** 10.3390/molecules28020843

**Published:** 2023-01-14

**Authors:** Jeffrey B. Krall, Parker J. Nichols, Morkos A. Henen, Quentin Vicens, Beat Vögeli

**Affiliations:** 1Department of Biochemistry and Molecular Genetics, University of Colorado Anschutz Medical Campus, Aurora, CO 80045, USA; 2Faculty of Pharmacy, Mansoura University, Mansoura 35516, Egypt; 3RNA Bioscience Initiative, University of Colorado Anschutz Medical Campus, Aurora, CO 80045, USA

**Keywords:** Z-RNA, Z-DNA, sequence preference, hydration, chemical modifications, junctions

## Abstract

Despite structural differences between the right-handed conformations of A-RNA and B-DNA, both nucleic acids adopt very similar, left-handed Z-conformations. In contrast to their structural similarities and sequence preferences, RNA and DNA exhibit differences in their ability to adopt the Z-conformation regarding their hydration shells, the chemical modifications that promote the Z-conformation, and the structure of junctions connecting them to right-handed segments. In this review, we highlight the structural and chemical properties of both Z-DNA and Z-RNA and delve into the potential factors that contribute to both their similarities and differences. While Z-DNA has been extensively studied, there is a gap of knowledge when it comes to Z-RNA. Where such information is lacking, we try and extend the principles of Z-DNA stability and formation to Z-RNA, considering the inherent differences of the nucleic acids.

## 1. Introduction

In 1979, when the first atomic structure of a macromolecular segment of DNA was solved [1], it came as a surprise that it adopted a left-handed, double helical structure instead of the model of B-DNA proposed by Watson and Crick decades before [2]. This newly discovered conformation of nucleic acids, termed Z-DNA, was nonetheless composed of two antiparallel strands and contained canonical Watson–Crick base pairs [1]. Soon after, it was found that polynucleotide RNAs were able to undergo the right-to-left-handed conformational change [3], and an early NMR solution structure of Z-RNA was modeled, indicating that a fold similar to Z-DNA is adopted [4]. While arguments of the biological relevance of Z-conformation nucleic acids were ongoing, proteins that could specifically recognize and bind Z-DNA were discovered [5,6,7], and were subsequently shown to also bind Z-RNA [8]. It was not until after this discovery that the first full Z-RNA structure was solved by NMR in high salt [9], and a low salt crystal structure in complex with Z-binding protein soon followed [10].

Since the 1990s, a plethora of biological studies have aimed to secure the relevance of Z-conformation nucleic acids in biological functions [11,12] (and references therein). Historically, the field as a whole has always been focused on the chemistry and biology of Z-DNA, leaving knowledge about Z-RNA to comparatively lag behind. Many experimental and theoretical studies that make up the bulk of knowledge about Z-DNA have not been extended or applied to Z-RNA. As a consequence, the chemistry and biology of Z-RNA is poorly characterized in comparison to that of Z-DNA.

In this review, we aim to discuss some of the inherent structural properties of Z-RNA and Z-DNA, and how they affect the Z-DNA/RNA formation, sequence preferences, stability, as well as energetics. While other reviews have focused more specifically on Z-DNA structure [13,14], the crystallization of Z-DNA sequences [15], Z-conformation biology [11,12,16,17,18], and the mechanisms of the B-to-Z transition [19,20], we focus in particular on extending our knowledge about Z-RNA formation and its relevance to biology.

## 2. Structural Characteristics of the Z-Conformation

### 2.1. General Properties of a Left-Handed Helix

The similarities between DNA and RNA in the Z-conformation come in contrast to the native states of the two nucleic acids, adopting the B- and A- conformation, respectively (Figure 1). Z-conformation nucleic acids adopt a narrower, more elongated double helix with helical rise and diameter of 44.6 Å and 18 Å, respectively, in contrast to both B-DNA (33.8 Å, 20 Å) and A-RNA (30.9 Å, 23 Å) (Table 1). In addition, the Z-conformation incorporates 12 bases per turn as opposed to the 10 and 11 in B-DNA and A-RNA, respectively [1,10,21,22,23]. A particularly striking feature of the Z-conformation is the zig-zag course that the phosphates follow along the backbone, from which the Z-form got its name (Figure 1, [1]).

### 2.2. The Z-Conformation Is Supported by Two Distinct Dinucleotide Steps

Although many sequences can adopt the Z-conformation [1,15,27,28,29,30,31,32], it is easiest to describe the conformational properties in reference to the (CpG)_3_ hexamer, the best characterized Z-conformation adopting sequence for both RNA and DNA [1,10]. As such, this section will use a d(CpG) repeat as the basis for describing the global properties of the Z-conformation because helical parameters for Z-RNA and Z-DNA are remarkably similar (Table 1) and deviations from a strict d(CpG) repeat will be discussed in detail later.

The unusual zig-zagging of the sugar-phosphate backbone is caused by alternating orientations of the ribose sugars in the guanosine and cytidine nucleotides [1]. In the Z-conformation, guanosine and cytidine residues have the furanose O4′ oxygen “pointing” in opposite directions along the phosphate backbone as in (Figure 2A [1,10]). To maintain left-handedness and base pairing, guanosine nucleobases are rotated 180° around the glycosidic bond while cytidine (nucleobase and ribose) is flipped upside down [1,10]. Rotation of guanosine’s glycosidic bond results in the adoption of the *syn* conformation and a concomitant adoption of a C3′-endo sugar pucker. Cytidine, however, remains in the *anti* conformation and adopts a C2′-endo pucker.

Alternating *anti*/*syn* conformations and C2′-endo/C3′-endo puckers within alternating pyrimidine/purine (APP) repeats results in a dinucleotide repeating unit [1], as opposed to the mononucleotide unit seen in A-RNA and B-DNA. Successive, overlapping dinucleotides adopt two distinct conformations, CpG (*anti*-p-*syn*; “Z-step” [34]) and GpC (*syn*-p-*anti*) (Figure 2A). For example, in the CpG step (Figure 2B), there is little rotation (−9°) between the nucleotides, whereas the GpC step (Figure 2C) has a much larger rotation (−51°). Together, the consecutive dinucleotides contribute to a total rotation of −60°, allowing for six units (i.e., 12 bases) to be incorporated in one full turn of the double helix (Table 1).

The two steps display different stacking and stabilizing interactions. In the CpG step (Figure 2a), a 7 Å base shear causes the six-membered ring of guanosine to stack upon the lone pair electrons of the O4′ oxygen of the preceding cytosine ribose moiety. This shear also results in the partial overlap of cytidine residues on opposing strands forming an inter-strand base stacking interaction (Figure 2B) [1,10]. The positioning of the lone pair electrons below the pyrimidine ring of guanine was originally thought to contribute to a stabilizing n_O4′_ → π* hyperconjugative effect [35]. However, Kruse et al. contradict this assessment as an artifact of suboptimal van der Waals (vdW) radii used in the original description [36]. Instead, they argue that the close lone-pair···pi (lp···π) contacts within the vdW depression of the pyrimidine ring enacts a minor, but acceptable, energy penalty as a result of the specific geometrical restraints of the Z-conformation [36]. The other lone pair of the O4′ oxygen projects perpendicularly to the helical axis, instead forming an intracytidine O4′ ··· H6–C6 hydrogen bond [37,38], probably through an n_O4_ → σ*_C6H_ hyperconjugation [35,39]. Because this interaction occurs within a single nucleotide, it is not specific to a single step.

The GpC step, in contrast, does not display significant base shearing, only has intrastrand base stacking effects, and does not display the lp···π contact (Figure 2C; [1,10,35,37,38,39]). Thus, the CpG (*anti*-p-*syn*) step contributes more favorable interactions to the stability of the Z-conformation as compared to the GpC (*syn*-p-*anti*) step and has been commonly referred to as the Z-step [34,40]. The differences in interactions between the two steps have been used to rationalize the ability of different oligo- and polynucleotides to adopt the Z-conformation when starting with the less favorable GpC step [41,42,43,44]. For short APP oligonucleotides starting with the less favorable *syn*-p-*anti* step, there is an uneven ratio of more favorable *anti*-p-*syn* to less favorable *syn*-p-*anti* steps (2:3 in the case of a (GC)_3_.). Thus, there generally needs to be a near even ratio of *anti*-p-*syn* to *syn*-p-*anti* steps to favorably adopt the Z-conformation, as seen in the polynucleotide d[GC]_n_ [42] and for d(GC)_n_ oligonucleotides where n ≥ 5 [43]. While the *anti*-p-*syn* “Z-step” has historically been seen as the defining dinucleotide of the Z-conformation [34], a recent structure of a left-handed quadruplex displays backbone conformations representing the “Z-like” *syn*-p-*anti* dinucleotide [45].

### 2.3. Phosphate Conformations Are Variable for the CpG Step

For nearly all structures solved to date, CpG phosphates exhibit the same conformation (Z_I_, Figure 3), characterized by phosphate oxygens OP1 and OP2 pointing in towards the minor groove [26]. However, GpC phosphates display more flexibility, and they can also adopt a Z_II_ conformation depending on the cations present in solution [26]. In the Z_II_ conformation, the phosphate is oriented away from the minor groove by ~1 Å towards the major groove surface, resulting in different patterns of hydration and allowing for the phosphate to contribute to direct ion binding interactions on the major groove surface [26,46]. The prevalence of the Z_I_ conformation and the restriction of the Z_II_ conformation to specific phosphates within crystal structures may suggest that the Z_I_ conformation is more energetically favorable [47]. The Z_II_ conformation has not been observed in any Z-RNA molecules of natural sequence [9,10]. The rotation of the GpC phosphate into a Z_II_ conformation, as seen in Z-DNA structures [33], would bring the OP1 oxygen within ~2.3 Å of guanosine’s 2′OH, and may contribute to unfavorable repulsive forces.

### 2.4. The Major and Minor Grooves of the Z-Conformation

The conformational properties of Z-form nucleotides result in a global helical structure that differs from that of A- or B-conformation nucleic acids and presents structural features unique to the Z-conformation (Figure 4A). The nucleobases of the Z-conformation are shifted away from the center of the structure, with the helical axis passing outside of the base pairs through the minor groove near O2 atoms of cytidine residues [1] (Figure 4B). Consequently, the minor groove is deep, extending 9 Å to the center of the double helix. In addition, the phosphate groups are closer together. The corresponding major groove, however, is everted in the Z-conformation due to the relationship of the bases to the phosphate backbone, brought on by the rotation of the nucleotides [13]. The rotations place the imidazole ring of guanosine nucleotides on the outer surface, exposing C8 and N7 and creating a convex surface instead of the typical groove seen in the A- and B-conformations [1,13]. N3 atoms of purines, which are common hydrogen bonding groups in both A- and B-conformation nucleic acids, are not typically accessible in the Z-conformation. Therefore, they act as another discriminating factor between the right- and left-handed conformations [15]. Otherwise, the grooves display the same potential hydrogen bonding surfaces as the B-conformation [15]. The implications of the conformational differences between Z, A, and B conformations are explored in later sections.

## 3. Formation and Stability of Z-conformations

### 3.1. Sequences That Can Easily Adopt the Z-conformation

Although any sequence is likely to have some ability to adopt the Z-conformation—albeit in a limited capacity [34]—the energy required to stabilize a longer sequence in a double helical Z-form depends on the nature of the base and on the context. For unmodified bases, a higher GC content is associated with a higher propensity to adopt a Z conformation [29,50,51,52,53]. APP sequences, such as the (CpG)_3_ described above, are more likely to adopt the Z-conformation due to lower energetic barriers as opposed to sequences that deviate from strict pyrimidine/purine repeats [13]. Base pair mismatches, such as G•T and G•Br^5^U wobble pairs, can also be accommodated within the Z-conformation with only minor perturbations to the overall structure as long as they maintain the APP sequence [54,55]. The major contributing factors as to why APP sequences with high GC content are preferred directly relates to the different abilities of nucleotides to adopt a *syn* conformation, as well as their ability to make stabilizing interactions through their substituent exocyclic groups and the subsequent solvent shells that are formed (reviewed in [56,57]). Although Z-DNA structures have been solved for various sequences, only one Z-RNA X-ray crystal structure of native sequence has been solved for the r(CpG)_3_ sequence [10], limiting comparisons that can be made between Z-DNA and Z-RNA structures of various sequences and their solvent shells.

In the following sections, we discuss factors contributing to the inherent stability of different Z-DNA sequences and how these might be extended to describe Z-RNA stability.

### 3.2. Purines Are More Likely to Flip to the Syn Conformation

The *anti* conformation is preferred in purines nucleotides due to having fewer steric clashes as compared to the *syn* conformation [26]. This is reflected in the conformational flexibility of purines, where the interconversion between the *syn* and *anti* conformations occurs at nanosecond timescales and the two states are approximately isoenergetic [58,59]. *Anti* to *syn* interconversions have been found to occur within DNA double helices, but the *syn* states are lowly-populated and short lived [60]. While these specific transitions were found to result in Hoogsteen base pairs, it may be a feasible nucleation event for Z-conformation adoption [60]. Pyrimidines, in contrast, are much less likely to adopt the *syn* conformation because the bulky O2 keto group would be located directly over the sugar moiety, resulting in close interatomic distances and unfavorable steric clashes [26,61]. Ultrasonic relaxation experiments support this claim, as no absorption can be measured for this interchange for pyrimidines, indicating that interconversion rates are much lower with higher energy barriers [58]. Ab initio modeling of the energy minima of *syn* and *anti* conformations and the energy barriers between them further concluded that, while all interconversion barriers ranged from ~3 to ~8 kcal/(mol∙bp), purines exhibited systematically lower energy barriers than pyrimidines [62,63].

The relative ease of purines to adopt a *syn* conformation helps explain why APP sequences are more likely to adopt the Z-conformation. To achieve the left-handed double helical structure, one base in a pair must be in *syn*, while the corresponding base on the other strand is *anti*. While many non-APP sequences have been crystalized and shown to adopt the Z-conformation in solution, the conditions (e.g., salt concentration) needed to facilitate the conversion are greatly increased compared to the CpG repeat, likely reflecting the increased energetic requirements [32,64,65,66]. For example, free-energy calculations of d(G•C) base pairs with cytidines and guanosines in the *anti* and *syn* conformations, respectively, required 0.3 kcal/(mol∙bp). However, out-of-alternation d(G•C) base pairs, with cytidines in *syn* and guanosines in *anti*, required 2.1 kcal/(mol∙bp) [65]. Pyrimidine bases placed in *syn*, regardless of nucleobase identity or context, are buckled out of the base pair plane and projected into the major groove [32,65,66], likely as a result of relieving steric constraints of the *syn* nucleotide [66]. This is not to say that non-APP sequences cannot adopt the Z-conformation. Non-APP sequences have been shown to adopt stable Z-conformation structures in solution using various conditions [17,66].

### 3.3. The Role of Hydration on the Stabilization of the Z-Conformation

#### 3.3.1. N2 Amino-Facilitated Water Networks Help Stabilize the Z-Conformation

Compared to pyrimidines, purines have relatively low energy barriers for the interconversion between *syn* and *anti* conformations. Among purines, guanine is more likely to adopt the *syn* conformation than adenine [26,58,59,62,63]. Theoretical work [67,68], ab initio modeling [62,63], and experimental studies [1,27,28,29,30,31,64,69] all suggest that the physical explanation for this increased stability is due to water-mediated hydrogen bonding networks facilitated by the exocyclic N2 group of guanosine residues, which is absent in adenosine residues. In the Z-conformation, N2 aminos of *syn* guanine bases form water-mediated hydrogen bond networks involving nearby phosphate groups (Figure 5A) [10,33]. Adenine, lacking an N2 amino group, is unable to make the same bridging connection and stabilize the *syn* conformation as effectively. Subsequently, a detailed analysis of the solvent shells of high resolution Z-DNA crystal structures with d(A•T) base pairs failed to see the well-defined electron density of water molecules in the minor groove typically associated with d(C•G) base pairs, indicating that solvent in this area was disordered (Figure 6A,C) [27,28,29,30,31]). While the solvation of d(A•T) base pairs was perturbed, the positioning of the nucleobases was not substantially different when compared to those of the d(G•C) base pairs.

Sequences with 2-aminoadenosine (2AA; Figure 6B) instead of adenosine have also been crystallized in the Z-conformation. The presence of an N2 amino group partially restores the water-mediated hydrogen bond network and has been theorized to lower the negative energetic penalties associated with A•T substitutions [71,72,73,74]. Loss of water organization appears to be partially sequence dependent, with some single A•T substitutions (Figure 6C) retaining water organization [31], whereas multiple, consecutive substitutions resulted in more severe disorganization [27,28,29,30]. The organization of water molecules around N2 aminos of guanosine residues and the phosphate backbone is similar in crystal structures of (CpG)_3_ Z-RNA and Z-DNA (Figure 5A), and the presence or absence of the N2 group is likely to affect the structure and stability of both nucleic acids in similar ways.

#### 3.3.2. O2 Groups of Pyrimidines Contribute to the Hydration of the Minor Groove

Well-ordered solvent shells exist around nucleic acids, and the interactions of water molecules with specific structural features of nucleic acids contribute to the overall stability of the molecule [75,76]. Well-defined interactions commonly occur around the phosphate backbone and across the major and minor grooves (reviewed in [56,57]). In addition to the N2 amino groups of guanosine residues, the exocyclic O2 groups of pyrimidines are another major stabilizing factor in the Z-conformation for APP sequences. Various crystal structures of different sequences have depicted a well-defined water molecule situated between adjacent pyrimidine bases of opposite strands (Figure 5B), in such a way that could facilitate hydrogen bonding between the O2 groups of successive pyrimidine residues [27,28,29,30,31,64,71,72]. These water molecules were thought to form a “spine of hydration”, as has been well quoted in the literature. However, recent cryo neutron crystallography results suggest that these waters form a different network [33]. Water molecules are actually oriented such that they are unlikely to connect successive O2 groups (Figure 7), instead forming a series of pentamers and hexamers that connect the interstrand phosphates across the minor groove [33]. Given the prevalence of the “spine of hydration” in Z-form nucleic acid literature, more cryo neutron crystallography studies are necessary to definitively rule out its presence. However, the presence of electron density for these specific water molecules is indicative of the stability of a given sequence because they are sensitive to disrupting factors [15]. In fact, disrupted spines are commonly seen in non-CpG APP sequences (Figure 6C,D) [29,30,72]; however, this has been speculated to be caused by differences on the major groove surface that propagate through the helix to widen or close the minor groove [15].

In addition to the water molecules mentioned above, an additional water molecule forms a hydrogen bond to the O2 atoms of cytidines and to a water molecule coordinated by the network formed by N2 aminos of guanosine and the phosphate backbone (Figure 5A) [13]. Thus, there are typically two waters in the plane of the base pair that are partially coordinated by both N2 and O2 atoms, creating a stabilizing pyrimidine-water-water-phosphate interaction [15], and together form the water pentamer structures mentioned above [33]. Similar to the spine of hydration, these water molecules are similarly affected by changes on the major groove surface and will be disordered for certain sequences or chemical modifications [15,29,30,65,72]. In Z-RNA, this network of water molecules is likely stabilized through an additional hydrogen bond due to the presence of a 2′OH and may contribute to the increased solvation generally seen in the minor groove of Z-RNA compared to Z-DNA (Figure 5A) [10].

#### 3.3.3. Solvation Patterns at the Major Groove Surface

There are two conserved patterns of hydration across the major groove surface (Figure 5C); however, they are typically only seen for CpG sequences because they directly link cytidine N4 to cytidine N4 and guanosine O6 to guanosine O6 of successive residues from one strand to the opposite strand [15]. In the first pattern, cytidine N4 amino atoms are bridged through two water molecules in the CpG step [15]. The orientation of water molecules in this pattern are variable, but the N4 groups always act as hydrogen bond donors [33]. Of the major groove hydration patterns, this appears to be the more stable of the two networks as it displays low B-factors and is not typically displaced by cations or polyamines [15], and is only slightly displaced by nearby chemical modifications [77,78].

The second water structure primarily occurs between the O6 atoms of guanosine residues in the GpC step (Figure 5C), and is bridged by one water between the two base pairs [15]. This bridging interaction is much less stable than the first pattern because it is easily displaced by cations and polyamines in crystals [15]. Because these interactions are specific to CpG repeats, substituting a single G•C base pair with A•T or A•U disrupts both patterns around the substitution (Figure 6C). Interestingly, replacing an entire CpG step with a UpA or Up2AA step leads to a similar network, but is facilitated through different functional groups (Figure 6B). The N4-water-water-N4 water pattern seen with CpG is replaced by an O4-water-water-O4 pattern, conferring a remarkably similar mode of hydration at this site, although the UpA waters are directly within the first shell of a hydrated magnesium ion [30,47]. The *anti*-p-*syn* hydration pattern is compensated in ApU and Ap2AA steps, while the *syn*-p-*anti* hydration patterns directly adjacent to the substituted step are lost due to incompatible exocyclic groups.

Solvation patterns of the major groove surface are similar between the Z-DNA and Z-RNA structures of the same CG sequence (Figure 5C). The 2′OH groups of pyrimidines face towards the minor groove where they stabilize the solvation of the minor groove in APP sequences [10]. The 2′OH groups of purines likewise face away from the major groove surface and likely do not interfere or directly contribute to the solvation of these two networks. Therefore, it is unlikely that there are major differences in solvation patterns across the major groove surface between Z-DNA and Z-RNA.

## 4. Chemical Conditions That Stabilize the Z-Conformation

Among the most studied conditions influencing the equilibrium between A-, B-, and Z-forms are the effects of chemical modifications (e.g. methylation, halogenation, etc.) [13,30,77,79,80,81,82,83,84], the effects of salts ([13,50,79,85] and references therein), and the effects of water activity (i.e. the effective concentration of water) through neutral solutes (methanol, ethanol, polyols, etc.) [15,39,86,87]. In contrast to what is known from studied sequence contexts, many chemical factors affecting the B-to-Z and A-to-Z transition differ between DNA and RNA. In the following section we will discuss select examples of chemical factors, excluding topological conditions and protein binding (for more information, see reviews [14,16,18] and references therein), and how they affect the equilibrium of the transition to the Z-conformation.

### 4.1. Covalent Modifications Affecting the Stability of the Z-Conformation

Covalent modifications of RNA and DNA nucleobases affect the stability of the Z-conformation by destabilizing the A- and B-conformations or stabilizing the Z-conformation. However, there is a disparity between how the same chemical modification affects RNA and DNA.

#### 4.1.1. Chemical Modifications at Pyrimidine C5 Promote Z-DNA, but Not Z-RNA

For Z-DNA, methylation at cytidine C5 destabilizes the B-conformation by rendering the major groove more hydrophobic while only partially destabilizing the Z-conformation, as the methyl group partially fills in a hydrophobic patch on the major groove surface and weakens the n_O4_ → σ*_C6H_ hyperconjugation [37,38,39,79]. The net result is the promotion of the Z-conformation (Figure 8). Methylation at C5 of Z-DNA results in minor perturbations to the Z-conformation structure. Methylated cytosines are pulled closer together, resulting in different helical twists for the d(m^5^CpG) steps and for the d(Gpm^5^C) steps (−13˚ and −46˚, respectively, [77]). The solvation patterns between adjacent O4 oxygens in the d(m^5^CpG) steps are not disturbed, but are slightly displaced due to the bulky methyl groups [77]. Unmethylated poly d[CG] only undergoes the B-to-Z transition at high salt concentrations (>2.5 M NaCl or >0.7 M MgCl_2_) [50], whereas these concentrations are drastically lowered for the methylated poly d[m^5^CG] (>0.7 M NaCl or >0.6 mM MgCl_2_) [79]. However, the same m^5^C chemical modification prevents the A-to-Z transition in RNA, requiring more salt to induce the transition in the poly r[m^5^CG] compared to poly r[CG] [19]. The direct cause of this destabilization is not entirely understood, as the methyl group is unlikely to sterically clash with or make unfavorable contacts with the preceding guanosines 2′OH [88].

The exocyclic methyl group of thymine at C5, in contrast, has little effect on the hydrophobicity of the major groove in B-DNA while increasing it for Z-DNA [15,51,73,91]. Likewise, higher salt concentrations are necessary to adopt the Z-conformation when d(A•T) base pairs are incorporated compared to the demethylated d(A•U) base pairs [30]. The methyl groups of thymine also destabilize the solvation patterns at the major groove surface of Z-DNA, whereas the same phenomena does not seem to occur with methylated deoxycytidine bases [29,30,47]. The d(TpA) steps showed a distinct lack of ordered water molecules at the major groove surface, especially around thymine N4, when electron density was analyzed [29]. In contrast, the C5 demethylated d(UpA) steps showed a restoration of this water bridging network facilitated through the outer sphere coordination of a hydrated magnesium cation with positions of waters similar to that seen in the CpG step (Figure 6A,D) [30].

Halogenation at the cytosine C5 position similarly stabilizes the Z-conformation of DNA, with iodination having a larger stabilizing effect than bromination [39]. The largely hydrophobic substitution destabilizes the B-conformation while promoting the adoption of the Z-conformation. However, unlike the sigma donating properties of methyl groups, the electron-withdrawing properties of halogens enhance the O4′ **^…^** H6–C6 hydrogen bond, contributing to further stabilization of the Z-conformation over methylation [37,38,39]. Similar to methylation, halogenation at C5 results in the cytosine bases being pulled closer together while simultaneously changing helical twists to a similar extent as methylation [78]. Halogenation also destabilizes Z-RNA similar to methylation, but this was originally proposed to be caused by the mutual repulsion of bulky electronegative halogens and the nearby 2′OH of 5′ guanosine riboses [88]. While this may partially explain the destabilizing effects of halogenation, it seems more likely that methylation and halogenation are acting through similar mechanisms.

#### 4.1.2. Any Modification at C8 Sterically Promotes the Z-Conformation

Unlike covalent modifications at C5 of pyrimidine residues, substitutions at the C8 position of guanosine nucleobases, and likely purines in general, promote the Z-conformation of both Z-DNA and Z-RNA (Figure 8) [13,88]. In both the A- and B-conformations, where the purines are in *anti*, the C8–H8 covalent bond points directly towards the phosphate backbone over the ribose moiety. Covalent modifications are generally not tolerated at C8 in the A- or B-conformation due to steric reasons. Early crystal structures indicated that 8-bromopurine nucleotides existed solely in the *syn* conformation due to close contacts between the bromo group and ribose moieties [92,93], and crystal structures of Z-DNA with methyl and bromo substitutions at C8 of guanosine nucleotides depict the substituent groups on the outside of the double helix away from any other functional groups [3,94].

Methylation at C8 of a single guanosine residue reduced the salt concentrations needed to induce the transition from 2.6 M to 30 mM NaCl [94], and molecular mechanic simulations of C8 bromination resulted in a reduction of free energy for the B-to-Z transition [88]. Methylation and bromination of C8 are likely to exhibit the same stabilizing effects for the Z-RNA structure. Modifications at C8 would destabilize the A-conformation while not affecting the stability of the Z-conformation, as implicated by molecular mechanics simulations [88]. In fact, incorporation of an m^8^G ribonucleotide into the middle step of a d(CG)_3_ hexamer resulted in the adoption of the Z-conformation without the addition of salts [94]. Non-specific bromination of poly r[CG], containing both Br^8^G and Br^5^C, leads to the adoption of the Z-conformation, indicating that the presence of a bromo group at C8 can stabilize Z-RNA even with the destabilizing bromo group at C5 [3].

### 4.2. Salts, Solvents, and Other Osmolytes

The most predictive method to analyze the stability of the Z-conformation in a specific sequence context or with a specific chemical modification is through the qualitative comparison of the concentrations of solutes needed to induce the A/B-to-Z transition [13]. This phenomenon has been known since before the first structure of Z-DNA was solved, and has been extensively studied since then [13,50]. However, the exact mechanisms that influence a solute’s ability to induce the transition to the Z-conformation still remain unclear and debated [39,86,95,96,97]. Many theoretical models trying to explain the rationale behind the solute-induced B-to-Z transition have arisen, including electrostatic theories [13,95,97], Hofmeister effects [15,96,98], direct ion binding [46,99,100,101], and osmotic pressure [15,39,86], or a combination of these effects.

#### 4.2.1. Electrostatic Contributions to the Adoption of a Z-Conformation

For salts promoting the Z-conformation, trivalent salts are better than divalent salts, which are in turn better than monovalent salts. Comprehensive models of electrostatic theories have been reviewed elsewhere [97], but generally, it is thought that cations of higher valencies are better able to screen the negative charges of the phosphate backbone and allow them to be spaced closer together. When the repulsive charges between phosphate backbones are neutralized, the Z-conformation is thought to be the preferred, lower-energy state [13,97]. The B-to-Z transition midpoints for a d(CG)_3_ occur at 2.7 M NaCl, 0.7 M MgCl_2_, or 30 μM cobalt hexamine ([Co(NH_3_)_6_]Cl_3_ [79]), indicating the increased effectiveness of higher valency cations in neutralizing phosphate charges and stabilizing DNA in the Z-conformation. Z-RNA is also more efficiently stabilized by cations of larger positive charge, but making direct comparisons between the A/B-to-Z transition is difficult due to (i) the relative inability of monovalent salts to induce the Z-conformation without also increasing the temperature [3], and (ii) the relative lack of midpoint concentrations for various salts promoting Z-RNA, which have been well characterized for DNA.

For the monovalent salts that can stabilize the Z-RNA transition, NaClO_4_ is the most well-studied and allows for a semi-qualitative comparison between the B-to-Z and A-to-Z transitions. The midpoint for the A-to-Z transition is at 4.09 M NaClO_4_ (10C; [102]), whereas a DNA hexamer with the same sequence undergoes the B-to-Z transition at 2.0 M NaClO_4_ (4C; [98]). Similarly, the A-to-Z transition for a r(CG)_3_ sequence requires higher divalent salt concentrations than the B-to-Z transition for poly d[CG], requiring 1.25 M MgCl_2_ (20C; [103]) and 0.7 M MgCl_2_ (~20C; [79]), respectively. Interestingly, cobalt hexammine has not been shown to stabilize Z-RNA although it does so efficiently for Z-DNA at micromolar concentrations [87]. Electrostatic screening was originally thought to be the governing factor for the stabilization of the Z-conformation [13], but generally fails to explain many of the trends of salts and the role of neutral solutes. In the following sections, we discuss other contributing factors leading to the promotion of the Z-conformation and the potential mechanisms behind them.

#### 4.2.2. Hofmeister Ions and Z-Conformation Stability

The Hofmeister series was first described in 1888 and ranked the ability of various ions to salt-in or salt-out proteins; however, the exact mechanisms behind the phenomenon were poorly understood at the time [104,105]. It is currently understood that ions in the Hofmeister series nonspecifically interact with macromolecular surfaces or their hydration spheres and influence hydrocarbon solubility, denaturation, surface tension, stability and other characteristics depending on their charge and hydrophobicity/hydrophilicity [105,106,107,108]. Ions in the direct Hofmeister series (Figure 9; left to right) have been known to promote the A- and B-to-Z transition, but this phenomenon cannot be solely explained by electrostatic screening [15,73,96]. If ions were stabilizing the Z-conformation purely through electrostatic screening, magnesium would be expected to be more efficient than calcium due to the smaller ionic radius of magnesium; however, calcium is slightly more efficient at promoting the B-to-Z transition [96]. Research into Hofmeister ions has largely been characterized in the context of proteins and other macromolecular systems [105,106,107,108,109,110,111], but less is known about their effects on nucleic acids, and key differences between them have been observed [112]. While Hofmeister effects have been noted for Z-DNA [73,96], caution should be taken when generalizations of the Hofmeister effects for proteins are applied to nucleic acids [112].

Specific anions have a more substantial effect on promoting the Z-conformation than cations do, as can be seen with NaCl and NaClO_4_ [3,4,15,39,79,86,87,98,102,103]. ClO_4_^−^ ions are more efficient at stabilizing the Z-conformation than Cl^−^ ions, as the midpoint transition only requires ~1.8M NaClO_4_ compared to ~2.5M NaCl for a d(CpG)_6_ sequence [96]. ClO_4_^−^ ions can more easily shed their solvent shells and adsorb onto macromolecular surfaces as compared to Cl^−^ ions [106,108,113]. In contrast, cations with higher charge densities, Mg^2+^ or Li^+^, will be more effective at neutralizing phosphate negative charges over ions of lower charge densities, such as Na^+^ or K^+^, as their transient charge–charge interactions are energetic enough to shed their hydrating waters [106,113].

Solvent free energy (SFE) calculations of Z- versus B-conformations indicate that Z-DNA is relatively more hydrophobic than its B-DNA counterpart (reviewed in [25,41]). Increasing concentrations of Hofmeister ions may therefore stabilize the more hydrophobic Z-conformation and facilitate the B-to-Z transition. Furthermore, APP repeats that are less stable in the Z-conformation than the prototypical d(CpG) repeat have more positive SFEs and may require larger salt concentrations [73]. Bulk FRET experiments have observed a strong correlation between B-DNA denaturation and B-to-Z transitions that similarly follows the direct Hofmeister series, which, along with other studies, indicates that partial denaturation or decreased helix stability of the B-conformation may be necessary for the structural change to the Z-conformation [96,114,115].

Ions in the Hofmeister series also decrease surface tensions and promote the adoption of macromolecular volumes with larger solvent accessible surface areas (SAS, [30,106]). SASs have been calculated for many dinucleotide repeats in the B- and Z-conformations [73], and show that the B-to-Z transition of d(CpG) dinucleotides results in an increase in SAS, whereas it decreases for d(TpA) dinucleotides [73]. CG-rich sequences require lower salt concentrations than AT-rich sequences to undergo the B-to-Z transition (reviewed in [15]), and the influence of ions to promote the adoption of larger SASs may partially explain the propensities of certain sequences in adopting the Z-conformation under high salt conditions. Many of the studies detailed here have not been replicated for A- and Z-RNA sequences, so it is difficult to definitively say whether the phenomena that governs Z-DNA adoption through the Hofmeister effect is the same for Z-RNA. However, ions following the direct Hofmeister series also stabilize RNA in the Z-conformation, albeit in higher concentrations, and might be acting through similar mechanisms [3,87,103].

#### 4.2.3. Examples of Direct Ion Binding

Another possible explanation for the increased effectiveness of divalent and trivalent salts in stabilizing the Z-conformation over monovalent salts is that divalent and trivalent cations may bind specifically to discrete locations on the Z-conformation structure (Figure 10). When considering these direct ion binding events, it is imperative to consider their relative effects on stabilizing the right- versus left-handed conformation. While many metals, ions, small molecules, and proteins have been shown to discretely bind the Z-conformation of nucleic acids, for the interest of brevity we will only detail a few examples of direct ion binding and its potential role in stabilizing the Z-form. Other reviews have detailed these direct interactions for Z-DNA [13,15,19,116]. Crystallographic analysis of cation binding sites on d(CG)_3_ sequences have found several discrete binding sites that magnesium and cobalt hexammine cations typically occupy [46,100]. Of interest, a fully hydrated magnesium cation coordinates the N6 and O6 functional groups of one guanosine and O6 of an adjacent guanosine on the opposite strand through its hydration shell (Figure 10A), resulting in three site-specific hydrogen bonds [46,117]. Crystal structures of the same hexamer in higher magnesium concentrations (500 mM) closer to the actual midpoint identified additional positions along the phosphate backbone and within the minor groove, with the latter making water mediated hydrogen bonds to OP1 and OP2 of both strands as well as with two consecutive cytosine O2 groups [100]. No crystal structures are available to locate direct binding sites of magnesium ions with Z-RNA and it is unclear if the presence of the 2′OH would exclude any of these magnesium binding sites. However, the fact that magnesium is still able to effectively promote the A-to-Z transition, albeit at higher concentrations (4.0 M MgCl_2_ [87]), may indicate that some of the binding sites might be similar, or instead, that MgCl_2_ is acting through separate mechanisms.

Binding sites for cobalt hexammine have also been described for Z-DNA [46]. The covalently attached amines of cobalt hexammine make three direct hydrogen bonds from the amino groups to O6 and N7 atoms of a single guanosine and two direct hydrogen bonds to the 5′ OP1 phosphate oxygen (Figure 10B) [46]. Thermodynamic studies agree with the direct binding model because titration of cobalt hexammine into d(CG)_4_ DNA octamers is an exothermic process, whereas titration of NaCl is largely endothermic. The authors concluded that this multivalent cation primarily stabilizes Z-DNA through site specific interactions whereas monovalent salts act through nonspecific interactions or through the dehydration of the native conformation, which will be discussed in the following section [119]. The stabilization of Z-DNA by cobalt hexammine may not be limited to CG repeats; a crystal structure of a d(CGCGCA)∙d(TGCGCG) shows a cobalt hexamine molecule directly binding to the A•T base pair. The binding of cobalt hexammine causes a tautomeric shift of the adenosine residue from the more prevalent amino tautomer into the imino form, the formation of a sheared A•T “wobble-like” pair, and the stabilization of the Z-conformation [120].

While cobalt hexammine greatly stabilizes Z-DNA, it lacks the same ability to stabilize Z-RNA [87]. The reason for this is poorly understood, but it may be due to the inability of RNA phosphates in the GpC step to adopt a Z_II_ conformation, as explained in Section 2.3. Without the ability of cytosine phosphates to flip into the Z_II_ conformation and the additional hydrogen bonds that the conformation provides, there may not be a discriminating surface between A- and Z-conformation nucleic acids to promote the A-to-Z transition. In both the high-salt NMR structure and the crystal structure, the phosphates of the Z-RNA oligonucleotide exclusively adopt a Z_I_-like structure [3,10]. However, both structures may have extraneous factors contributing to the sole adoption of a Z_I_ phosphate conformation. The NMR solution structure was solved in 6M NaClO_4_ and the structure likely represents a dehydrated form of the Z-RNA molecule [3,9,10], whereas the crystal structure employed a Z-conformation binding protein, Zα, to crystallize the nucleotide, and restraints induced by the protein may prevent the adoption of a Z_II_ conformation typically induced by transition metals and cations [1,10,13,15,46,100,121].

#### 4.2.4. Promotion of the Z-Conformation through Osmotic Pressure

It has been proposed that preferential hydration of solutes over polynucleotides or the exclusion of solute-inaccessible pockets in macromolecules may drive the B-to-Z and A-to-Z transitions, and may explain why the A-to-Z transition requires higher concentrations of dehydrating agents over the B-to-Z transition [15,39,86]. The B-to-Z transition has been shown to be largely entropically driven, with the main contributing factor being the release of waters due to differences in hydration between the two conformations [119]. The addition of salts or neutral osmolytes would decrease the water activity and induce an osmotic pressure that would shift the equilibrium to favor a conformation that reduces hydrating waters or solute-excluded volumes (Figure 11) [15,19,119,122]. The transition from the B- to Z-conformation results in the release of ~2.5–7.7 water molecules per base pair, depending on the experimental conditions and calculations used [19,119].

The addition of neutral solutes that are not expected to directly interact with nucleic acids, such as the zwitterion betaine [119], monohydroxyl alcohols such as methanol, ethanol, or propanol, and polyols such as sucrose, stachyose, and many others [19,86,87], promote both the B- and A-to-Z transitions. Osmotic pressure theories also help explain why the A-to-Z transition occurs at greater concentrations of osmolytes than the B-to-Z transition. The A-conformation of DNA is similarly preferred at low water activity and humidity [123,124,125]. In the A- and Z-conformations of DNA, the hydration shells of the phosphates overlap, whereas they are isolated in the B-conformation [78,126]. The increased efficiency of the hydration of phosphates in the A- and Z-conformations may explain why they are favored at low water activities [15,57,124,127]. For A-RNA, the thermal stability of the duplex has been attributed to the increased stability of water networks in the major and minor grooves due to the presence of 2′ hydroxyl groups that act as a scaffold and rigidify the A-conformation [127]. Though the hydration of A-RNA requires more water molecules than B-DNA does [127], the increased stability of this water network and the efficiently hydrated phosphate groups may require larger osmotic pressures to disrupt.

## 5. Energetics of Z-Formation

### 5.1. Inherent Differences of Sugar Puckers for Ribose and Deoxyribose

The most significant conformational differences between dsDNA and dsRNA are related to the sugar pucker of the ribose sugars [26]. Nucleotides typically adopt either a C2′- or C3′-endo sugar pucker as both correspond to local energy minima across the wide variety of potential pucker conformations and can interconvert between them with relative ease [26,128]. For both RNA and DNA, the C2′-endo and C3′-endo energy minima are relatively close to each other, with RNA preferring the latter over the former (~2 kcal/mol difference) and DNA preferring the former over the latter (~1 kcal/mol difference) [128]. The lowest energy barrier between the two states (proceeding through an O4′-endo pucker), is approximately 4.5 kcal/mol and 5.0 kcal/mol for deoxyriboses and riboses, respectively, as determined by force-field calculations and falls within the range of experimentally determined values [129]. The only chemical difference affecting conformational properties of riboses in RNA and DNA stems from the presence of a 2′OH [26,128], and its presence contributes to different steric and stereoelectronic effects, as well as different potential hydrogen bonding patterns, that alter the intrinsic energies and the conformational dynamics of the ribose puckers [129,130,131,132,133,134].

With these energy differences in mind, it is important to remember that Z-RNA and Z-DNA adopt remarkably similar conformations, and as highlighted in Section 2, successive nucleotides alternate between *syn*/C3′-endo and *anti*/C2′-endo conformations [1,10]. For DNA, where the C3′-endo and C2′-endo puckers are similar in energies (~1 kcal/mol different) and the energy barrier between them is lower (4.5 kcal/mol), the energetic penalty for purines to adopt the less favorable C3′-endo conformation would also be lower and occur with a higher frequency [129]. However, for RNA, where the energy differences between the two puckers is larger (~2 kal/mol different) and the energy barrier is higher (5 kcal/mol), the energetic penalty for pyrimidines to adopt the less favorable C2′-endo conformation would be higher and proceed with more difficulty than a similar interconversion in DNA [129].

### 5.2. Energetics of the A-to-Z and B-to-Z Transitions

Thus, it takes more energy to alter the pucker in dsRNA than dsDNA (Figure 12), and consequently, the A-to-Z transition for r(CpG)_6_ sequences is associated with higher activation energies (~3.2 kcal/(mol∙bp)) compared to the B-to-Z transition for the same DNA sequence (2 kcal/(mol∙bp)) [8]. For chemically induced transitions of (CpG)_n_ sequences, the transition enthalpies differ depending on the inducers, techniques, and chain length used in the studies, but all agree that the conformational change occurs with positive transition enthalpies for both RNA and DNA [50,119,135,136,137]. Interestingly, calculations of internal thermodynamic forces using harmonic approximation show that an entropic penalty is associated with switching from the B- to Z-form as a consequence of the increased rigidity of the Z- over B-conformation [138], and agrees well with previous values obtained from NMR measurements [139]. However, the decrease in entropy as a consequence of backbone rigidity is outweighed by an overall gain of entropy as a result of the release of waters accompanying the conformational change induced by monovalent salts [50]. As has been discussed in the previous section, dehydrating conditions favor the transition to the Z-conformation [19,119], and the consequent release of water provides an entropic gain that promotes the conformational switch.

The impact of the differences in energy barriers between the two states can be readily seen by following B- and A-to-Z transitions in d/r(CpG)_6_ repeats upon binding to the Zα domain using circular dichroism, which showed that Z-RNA adoption is much slower compared to Z-DNA for the same temperature [8]. Only once reaching 50 °C did Z-RNA achieve a comparable rate to Z-DNA at 25 °C [8]. Further illustrating this point, induction of Z-DNA in d/r(CpG)_n_ repeats requires significantly less ionic strength (2.7 M NaCl) than the formation of Z-RNA (6 M NaClO_4_) [3,50].

### 5.3. Energies of Z-Formation Depends on Sequence Context

The free energies of A-to-Z and B-to-Z transitions also heavily depends upon the sequence context of a given nucleic acid. There have been numerous studies and reviews detailing the ability of a given DNA dinucleotide to adopt the Z-conformation, and the general series can be observed from experimental data: d(m^5^CG) > d(CG) > d(CA)-d(TG) > d(CC)-d(GG) > d(CT)-d(GA) > d(TA) ([15,19] and references therein). The energies for the dinucleotides range from −0.6 to 2.4 kcal/(mol∙dn), and heavily depend on the experimental conditions [15]. Free energies of adopting the Z-conformation have been calculated for other dinucleotides (including d(U2AA), d(T2AA), d(UA), etc.), but many have not been experimentally verified to date [15]. These differences have largely been attributed to base-stacking interactions, sterics, and hydration [15]. The bulk of research has been solely focused on the energetics of dinucleotides for Z-DNA but have been largely neglected for Z-RNA. Outside of the prototypical CG sequences, there is still much to learn about the free energies of the A-to-Z transition for different RNA sequences.

## 6. Junctions between Right- and Left-Handed Segments

### 6.1. Z-DNA and Z-RNA Adopt B-Z, A-Z, and Z-Z Junctions

For biologically relevant double-stranded nucleic acids, the majority of bases adopt the low-energy A- or B-conformations; however, it has been experimentally suggested that the Z-conformation is adopted in both DNA and RNA under the appropriate physiological conditions [140,141,142,143]. For a segment of RNA or DNA to flip into the Z-conformation within a larger stretch of A- or B-conformation nucleic acid, the handedness of the helix must reverse twice, creating two B-Z or A-Z junctions for DNA and RNA (Figure 13), respectively [144,145,146,147]. In the first crystal structure of a B-Z junction (Figure 13B), the helicity reversal was accomplished by extruding an A•T base pair between the B- and Z-forming regions while simultaneously maintaining tight base stacking interactions between the two conformations [144]. The extrusion of an A•T base pair is a common feature of junctions and is used extensively to monitor their formation; however, it is not necessary to facilitate junction formation under different chemical inducers [148].

While B-Z junctions have been well characterized for DNA sequences both structurally and biophysically, a structure for an A-Z junction has not been solved to date (Figure 13A). Biophysical studies suggest that B-Z and A-Z junctions behave differently, with A-Z junctions exhibiting partially melted base pairs or poor base stacking interactions extending beyond the extruded base [145,146]. The differences in base stacking environments are larger between A- and Z-conformations than those between B- and Z- conformations, including helical diameters, base tilts, and positioning of the helical axis (Figure 13A, Table 1), and may explain the loose stacking seen in A-Z junctions [145].

### 6.2. A-Z and B-Z Junctions Contribute Energetically to Z-Conformation Adoption

In the context of relevant double-stranded polynucleotides, Z-forming regions are embedded in longer stretches of A- or B-conformation helices. Thus, it is also important to consider the thermodynamics of A-Z or B-Z junctions, as two junctions must be formed in order to accommodate a left-handed segment and can be a driving force for Z-conformation adoption [145,150,151]. The adoption of a B-Z junction has only a minor effect on the overall stability of the DNA duplex. Using increasing salt concentrations to promote the formation of a B-Z junction, the free energy of melting the duplex only decreased by ~0.5 kcal/mol compared to a right-handed control [60].

Using the Zα domain, FRET studies showed that the primary thermodynamic driving force for Z-formation is the favorable increase in entropy associated with forming the junctions between a Z-DNA/Z-RNA stretch and the adjacent B- and A-forming regions (Figure 14). This effect is more favorable for RNA A-Z junctions than for DNA B-Z junctions, with A-Z junction formation occurring more quickly [151]. This can be attributed to either the increased entropic gain resulting from exposing an RNA nucleotide to the solvent compared to DNA, or the exposure of multiple nucleotides. This conformational change helps to overcome the unfavorable energetics associated with riboses adopting a C2′-endo pucker, which would normally hinder Z-RNA formation [151].

It was also found that DNA/RNA hybrids (where an RNA strand is paired with a DNA strand) are the best junction-adopting sequences due to combining the most favorable thermodynamic properties of junction formation for DNA and RNA [151]. This is because the hybrid takes the more favorable enthalpic contribution of flipping the d(CpG) sequence to the Z-form from DNA and the more favorable entropic contribution of forming the junction from RNA, resulting in the lowest overall free energy of the tested constructs [151]. Overall, these results emphasize that the energetics of Z-DNA and Z-RNA depend heavily on positional context within larger stretches of non-Z-conformation nucleic acids.

## 7. Outlook

DNA and RNA are unique from a chemical, geometrical, and structural perspective (Figure 1 and Table 1) [128,152]. Despite having different ground state conformations, both DNA and RNA can adopt the higher energy left-handed Z-conformation. Z-DNA and Z-RNA are similar from a structural standpoint [4,153,154], yet they display different thermodynamics, conditions that promote stabilization, and junction formation. The forces driving Z-DNA formation have been relatively well characterized both structurally and biophysically. In contrast, there have only been two deposited Z-RNA structures of the same (CG)_3_ sequence, making structural comparisons difficult [4,10]. The first was solved in the presence of high salt (~6.0M NaCl), likely depicting a highly dehydrated conformation [4], and the other in the presence of a stabilizing protein [10]. In order to better understand the inherent differences between Z-DNA and Z-RNA and be able to make direct comparisons between them, more structures of Z-RNA with different sequence contexts should be undertaken.

While the chemical conditions and modifications driving the adoption of the Z-conformation is relatively understood for Z-DNA [13,15,30,40,50,77,79,80,81,82,84,85,86], much less is known for Z-RNA [83,87]. Some of the same chemical modifications that promote DNA’s adoption of the Z-conformation prevent it for RNA [87]. While it can be speculated that these differences are caused by the structural differences of the A- and B-conformation, it has yet to be proven. Similarly, the different effect of salts and osmolytes on Z-RNA and Z-DNA adoption could be due to a number of factors including differences in hydration of the A- and B-conformations [127], or accessibility of Z_I_ or Z_II_ phosphate positions to coordinate specific salts. Ions in the Hofmeister series have been known to promote the Z-conformation [73,96]; however, the exact mechanisms behind this phenomenon have not been described for either Z-DNA or Z-RNA and remains poorly understood for nucleic acids in general [112].

Finally, little is known about RNA A-Z junctions in comparison to DNA B-Z junctions, although such junctions are likely an indispensable determinant in the cellular context. While several structures of B-Z junctions have been deposited [144,147], to date, a structure of an A-Z junction has not been solved [146]. There is an indication that A-Z junctions behave differently compared to B-Z junctions and may be larger than one base pair. While this may be explained due to the larger difference between the helical axis of A- and Z-conformation nucleic acids, this needs further investigation.

The physiological importance of Z-conformation nucleic acids is still an ongoing debate, even after more than four decades of study [18]. While both Z-DNA and Z-RNA have been implicated in separate biological processes, concrete proof of their existence in vivo remains elusive and/or difficult to detect [13,14,15,18,155]. As DNA and RNA exist in drastically different contexts within the cell, it becomes imperative to understand the conditions that may stabilize the Z-conformation for each in their respective environments.

## Figures and Tables

**Figure 1 molecules-28-00843-f001:**
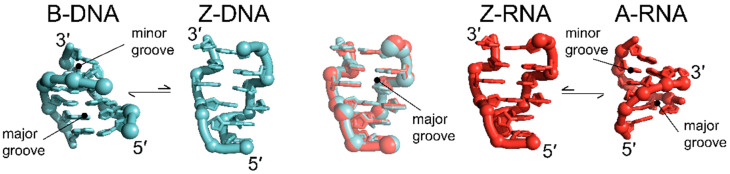
Z-DNA and Z-RNA are structurally equivalent. Although the right-handed B- (left, cyan, PDB: 1N1K [24]) and A-conformations (right, red, PDB: 1PBM [25]) of helical DNA and RNA are structurally distinct from each other, they can both adopt the left-handed Z-conformation (middle, PDBs: 1QBJ [7] for DNA and 2GXB [10] for RNA). Unlike the phosphate backbones of A- and B-conformation nucleic acids, which follow a smooth curve, the backbone of the Z-conformation zig-zags. Overall, the helical parameters of Z-DNA and Z-RNA are very similar (Table 1), with the opening of the minor groove of Z-RNA only being marginally smaller than that of Z-DNA.

**Figure 2 molecules-28-00843-f002:**
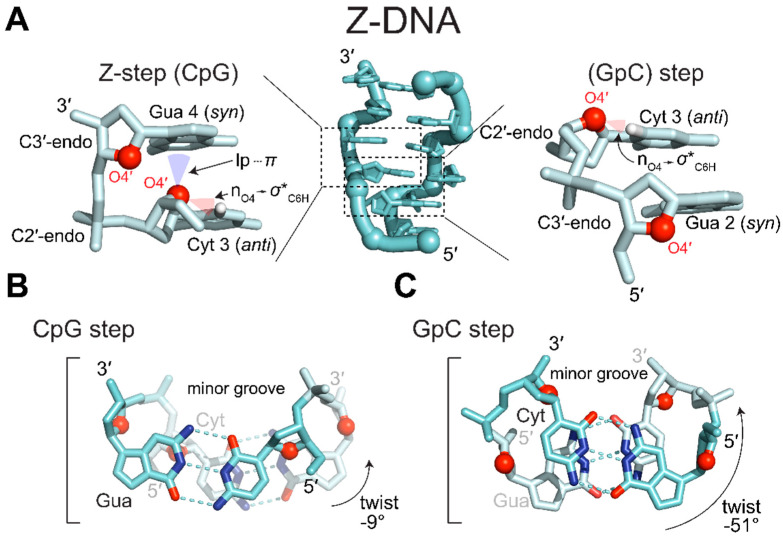
The building block of the Z-conformations are two distinct dinucleotide steps. (**A**) d(CpG)_3_ sequence in the Z-conformation (PDB: 7JY2 [33]). Z-form nucleic acids are composed of two distinct, overlapping dinucleotides steps. In both steps, ribose O4′ (red spheres) alternate between “pointing up” and “pointing down”. Pyrimidines adopt *anti* conformations and C3′-endo puckers, whereas purines adopt *syn* conformations and C2′-endo puckers. On the left, the Z-step (*anti*-p-*syn*) exhibits a unique contact not typically seen in other steps or helical conformations. One lone pair of oxygen O4′ sits within the vdW depression of the guanidinium system, resulting in lp···π contacts and close nucleobase plane distances. The other lone pair of oxygen O4′ facilitates an n_O4_ → σ*_C6H_ hyperconjugation. The other step (*syn*-p-*anti*) does not include the lp···π contacts but does include the n_O4_ → σ*_C6H_ hyperconjugation. (**B**) Top-down view of the CpG step. O4′ can be seen pointing into guanine’s six-membered ring close to C2 and the interstrand cytidine base stacking can also be seen. (**C**) Top-down view of the GpC step. There is no lp···π contact and only intrastrand base-stacking occurs.

**Figure 3 molecules-28-00843-f003:**
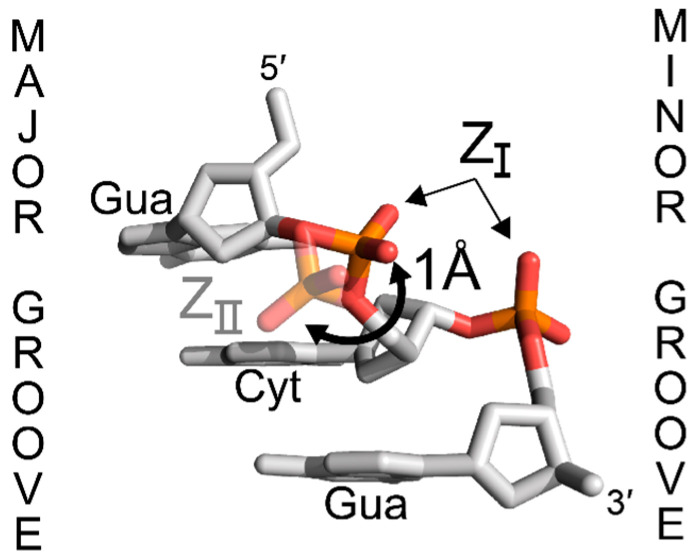
*Syn*-p-*anti* steps can adopt two phosphate conformations in Z-DNA. While all *anti*-p-*syn* steps adopt a nearly uniform Z_I_ phosphate position, *syn*-p-*anti* are more flexible and can adopt two distinct phosphate positions, Z_I_ or Z_II_. Z_I_ is more prevalent and is characterized by the phosphate pointing in towards the minor groove. In Z_II_, the phosphate is rotated away from the minor groove by 1 Å towards the major groove surface. It is unclear if Z-RNA can adopt a similar Z_II_ position. Figure adapted from PDB: 7JY2 [33].

**Figure 4 molecules-28-00843-f004:**
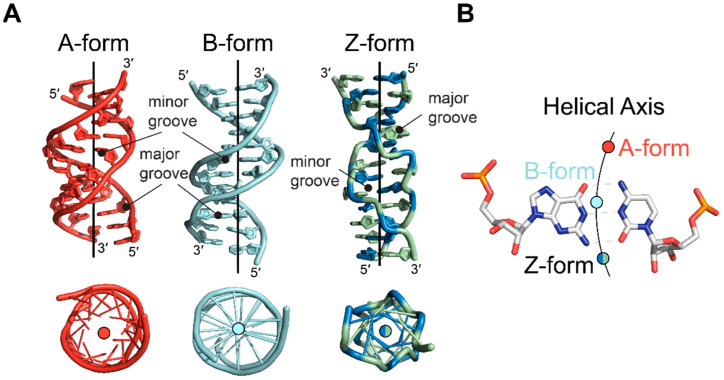
Comparison of helical axes between A-, B-, and Z-form nucleic acids. (**A**) Side (**top**) and top-down (**bottom**) views of A-form (red, PDB: 413D [48]), B-form (cyan, PDB: 6CQ3), and Z-form (green and blue, PDB: 4OCB [49]) nucleic acids. Position of helical axes are drawn as a line for each structure. Below, the structures have been rotated to look down the center of the helix. (**B**) Positions of the helical axes relative to a G•C base pair for the A-form (red circle), B-form (cyan circle), and Z-form (green and blue circle) helices.

**Figure 5 molecules-28-00843-f005:**
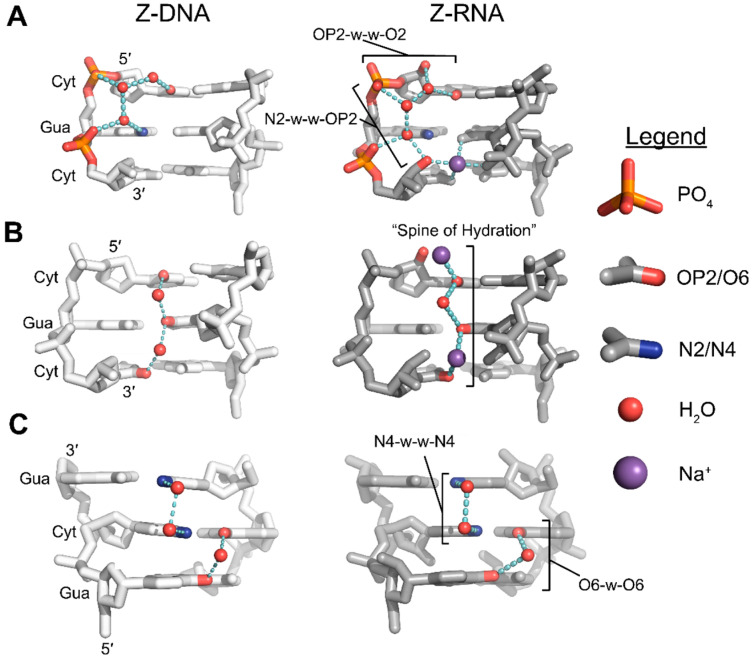
Hydration of the Z-conformation plays a significant role in its stabilization. Hydrogen bonding networks facilitated by exocyclic groups of alternating purines and pyrimidines to the phosphate backbone within the minor groove in Z-DNA (PDB: 1ICK [70], **left A**,**B**,**C**) and Z-RNA (PDB: 2GXB [10], **right A**,**B**,**C**). (**A**) Guanine N2 amino groups are involved in hydrogen bonding networks to both the 5′ phosphate of the following residue (N2-w-OP2) or to its 5′ phosphate (N2-w-w-OP2). Cytidine O2 groups are involved in hydrogen bonding networks to 5′ phosphates of the following residue through two water molecules (O2-w-w-OP2) or to guanine N2 groups through one water (O2-w-N2). In Z-RNA, the 2′OH is within hydrogen bonding distance to stabilize the networks through additional hydrogen bonds. (**B**) Spine of hydration for Z-DNA (**left**) and Z-RNA (**right**). Water molecules are positioned between the base plane and were thought to facilitate hydrogen bonds between successive cytidine residues of opposite strands. In the Z-RNA crystal structure, every other water molecule is replaced by Na^+^, reducing H-bond length and resulting in a narrowing of the minor groove. (**C**) Hydration of the major groove surface for Z-DNA (**left**) and Z-RNA (**right**). Two water molecules bridge N4 groups of adjacent cytosine residues on opposite strands. One water molecule bridges O6 atoms of adjacent cytosine residues on opposite strands.

**Figure 6 molecules-28-00843-f006:**
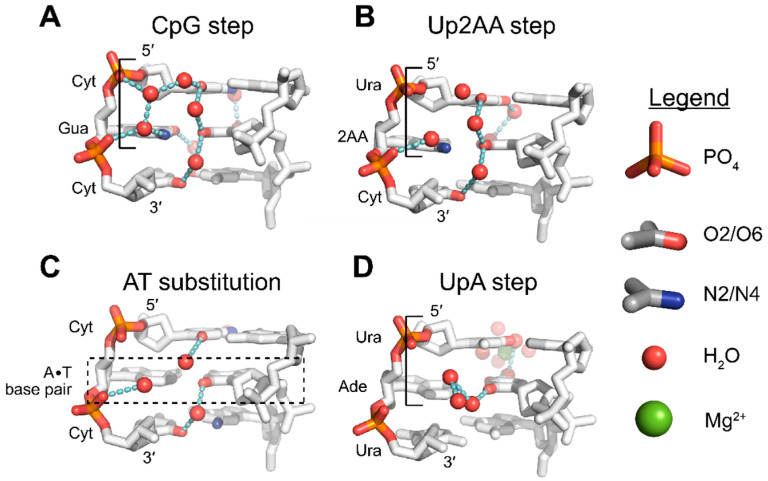
Hydration patterns of non-CG Z-steps as a comparison to the CpG step. Hydration patterns around different DNA sequences as they compare to the d(CpG) step. Water molecules are colored red with dashed lines representing hydrogen bonding networks. (**A**) Common hydration patterns of the reference d(CpG) step (PDB: 1ICK [70]) as mentioned above. (**B**) Presence of N2 substituent group in the d(Up2AA) step (PDB: 1D76 [47]) partially maintains hydration within the minor groove. On the major groove surface, consecutive O4 atoms are linked by similar water-bridging interactions seen between consecutive N4 atoms in the CpG step; however, the water bridging the O6 atoms of the adjacent dinucleotide step is absent. (**C**) Substitution of a single G•C base pair with A•T (shown within the dashed box), results in a CA·TG step and only partially disrupts hydration (PDB: 181D [31]). (**D**) Demethylation of dT bases to dU in d(UpA) steps (PDB: 1D41 [30]) results in the reordering of water molecules within the minor groove, but the hydration pattern is different from what is typically seen in d(CpG) sequences. Outer sphere coordination of a magnesium cation facilitates the two-water bridge between consecutive O4 atoms.

**Figure 7 molecules-28-00843-f007:**
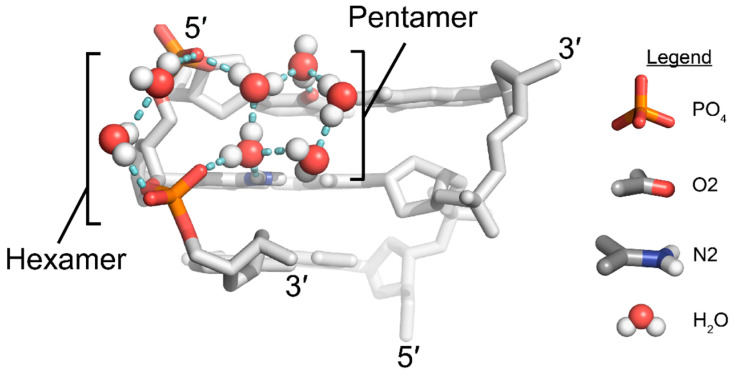
The minor groove is hydrated by hexameric and pentameric water structures. Position and orientation of water molecules in a d(CG)_3_ structure, as seen by cryo neutron crystallography (PDB: 7JY2 [33]). Similar water networks are observed as those in Figure 5; however, no spine of hydration is possible given water orientations. Hydration of the phosphates and exocyclic groups results in hexameric and pentameric water structures that are mirrored across the minor groove. It is unknown if these water structures are similar in Z-RNA.

**Figure 8 molecules-28-00843-f008:**
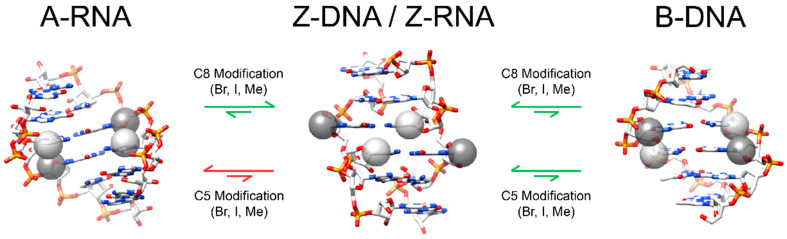
Chemical modifications promote or hinder Z-form adoption in nucleic acids. Structures of six-nucleotide segments of A-RNA (PDB: 4MS9 [89]), Z-RNA (PDB: 2GXB [10]), and B-DNA (PDB: 2M2C [90]) with central G•C base pairs are shown. The positions of pyrimidine C5 or purine C8 chemical modifications are modeled as light grey or dark grey spheres, respectively. In both A-RNA and B-DNA, modifications at cytidine C5 or guanosine C8 result in the chemical groups close to the phosphate with C8 often sterically clashing with phosphate groups. In the Z-conformation, C5 modifications are placed in the center of the major groove surface and C8 modifications are on the edges pointing out towards solvent. The reason for why C5 modifications hinder the A-to-Z transition is poorly understood for both methylation and halogenation.

**Figure 9 molecules-28-00843-f009:**
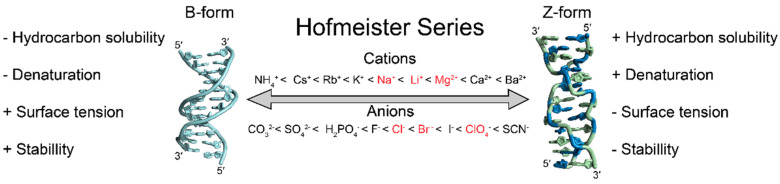
Hofmeister salts stabilize the Z-conformation. Hofmeister salts influence macromolecular structures by affecting hydrocarbon solubility, denaturation, surface tension, and stability [105,106,107,108,109,110,111]. The B-to-Z transition has been shown to be promoted by Hofmeister salts, although the direct cause has not been directly studied. Contributing factors may be due to differences in solvent accessible surface area or hydrophobicity between the right- and left-handed conformations or by promoting the Z-conformation through destabilization of the right-handed conformation. Ions highlighted in red can be definitively placed for salts affecting the A-to-Z transition as well [3,87,103].

**Figure 10 molecules-28-00843-f010:**
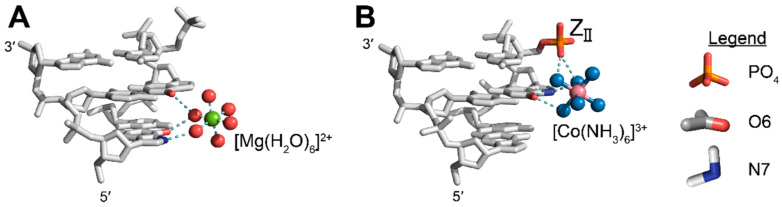
Examples of direct ion binding to Z-DNA. (**A**) A hydrated magnesium binds to the major groove surface of a d(CpG) step, forming three hydrogen bonds between two interstrand guanines (PDB: 1DCG [117]). Guanine O6 and N7 of one strand and guanine O6 of the opposite strand are bridged. (**B**) Cobalt hexammine directly binds to one strand of a d(CpG) sequence, forming 5 hydrogen bonds (PDB: 331D [118]). Two N2 groups hydrogen bond to OP2 of a Z_II_ phosphate, two N2 groups hydrogen bond to guanine O6, and another N2 group hydrogen bonds to N7 of the same guanine. Cobalt hexammine stabilization appears to be specific to Z-DNA and not Z-RNA.

**Figure 11 molecules-28-00843-f011:**
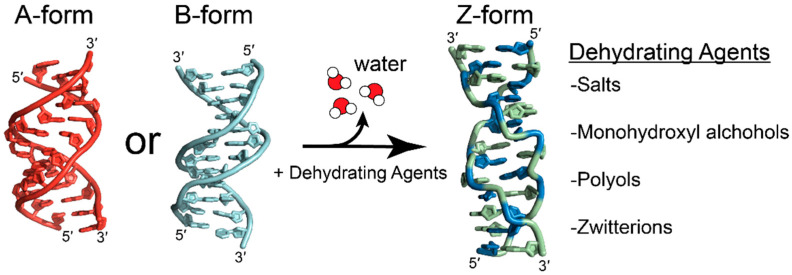
Dehydrating conditions favor the Z-conformation. Addition of dehydrating chemicals, such as salts, alcohols, and osmolytes favors the Z-conformation (green and blue, PDB: 4OCB [49]) over the right-handed A-RNA (red; PDB: 413D [48]) or B-DNA (cyan, PDB: 6CQ3) conformations due to the increased efficiency of hydrating the closely spaced phosphate backbones in the Z-conformation.

**Figure 12 molecules-28-00843-f012:**
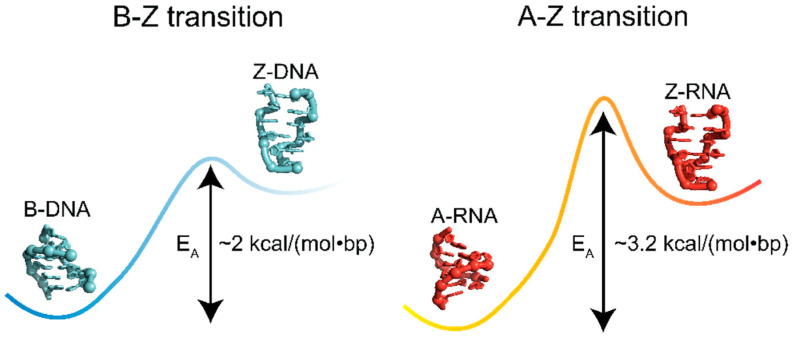
The B-to-Z transition is lower in energy compared to the A-to-Z transition. The activation energy for the transition from B-DNA (PDB: 1N1K [24]) to Z-DNA (PDB: 1QBJ [7]) is ~1.2 kcal/(mol∙bp) lower than the transition from A-RNA (PDB: 1PBM [25]) to Z-RNA (PDB: 2GXB [10]).

**Figure 13 molecules-28-00843-f013:**
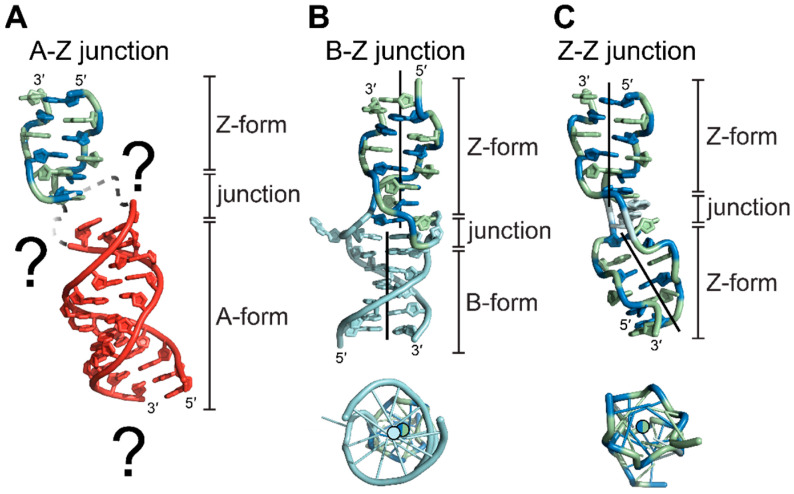
Structures and features of A-Z, B-Z, and Z-Z junctions. Side (**top**) and top-down (**bottom**) views of A-Z junctions (adopted from PDBs 2GXB and 413D for the Z- and A-segments, respectively, [10,48]) B-Z junctions (PDB: 2ACJ [144]), and Z-Z junctions (PDB: 3IRQ [149]). (**A**) The structure of an A-Z junction has not been solved, although characterizations of A-Z junctions suggest they may extend beyond one base pair. The difference in the position of the helical axis is much greater between A- and Z-conformations than it is for B- and Z-conformations (Figure 4), which may partially explain the need for a larger junction segment. (**B**) B-Z junctions facilitate the continuation of base stacking between segments of B- and Z-conformation nucleic acids and can include extruded bases, although this is not always the case. The helical axis between the two segments (solid lines) are only slightly displaced and largely parallel to each other. (**C**) Junction formation is possible between two segments of Z-conformations, excluding a potentially unfavorable nucleotide from adopting the Z-conformation while continuing stacking interactions. The introduction of a junction between two segments of Z-conformation nucleic acids kinks the helical axis.

**Figure 14 molecules-28-00843-f014:**
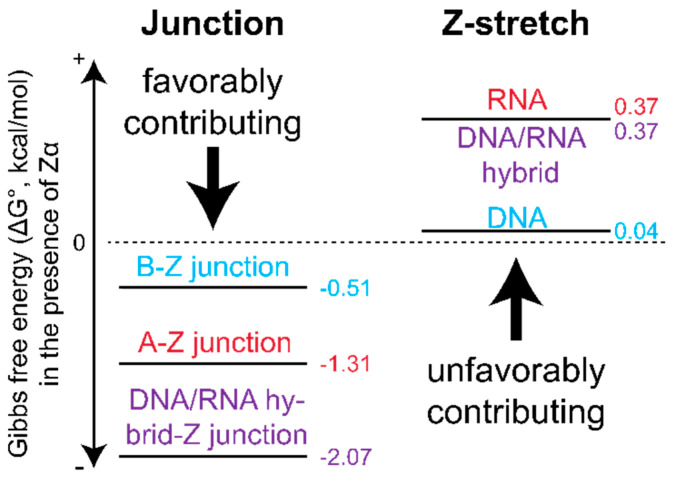
Energetics of junction formation. Junction formation dominates free energies of Z-conformation adoption and largely drives the transition in the presence of protein. DNA/RNA hybrid junction formation is the most energetically favorable, followed by A-Z junctions, and then B-Z junctions. CpG Z-formation is unfavorable in every case but is lowest for DNA and equivalent for RNA and DNA/RNA hybrids. All ΔG values reported from [151].

**Table 1 molecules-28-00843-t001:** Helical parameters for nucleic acids in their left- and right-handed conformations.

Helix Sense	B-DNA ^1^	Z-DNA (Z_1_) ^1^	A-RNA ^1^	Z-RNA (Z_D_) ^2^
	Right-Handed	Left-Handed	Right-Handed	Left-Handed
Residues per Turn	10	12	11	12
Pitch Height (Å)	33.8	44.6	30.9	44.6
Diameter (Å)	20	18	23	18
Rise per Residue (Å) ^3^	3.38	3.7	2.81	3.7
Base Tilt (˚)	−6	−7	13	−7
Rotation per Residue (˚) ^4^	36	−9, −51	32.7	−8.6, −50.9
Nucleoside Conformation				
Guanosine	*Anti*	*Syn*	*Anti*	*Syn*
Cytidine	*Anti*	*Anti*	*Anti*	*Anti*
Sugar Pucker				
Guanosine	C2′-endo	C3′-endo	C3′-endo	C3′-endo
Cytidine	C2′-endo	C2′-endo	C3′-endo	C2′-endo

^1^ B-DNA, Z-DNA, and A-RNA helical parameters are adapted from [26]; ^2^ Z-RNA average helical parameters from the crystal structure (2GXB) are reported; ^3^ Rise per Residue is averaged between the two distinct steps for the Z-conformations; ^4^ The two values reported for Z-conformation nucleic acids are for the CpG and GpC steps, respectively.

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
