# Peer review of "Structure and Formation of Z-DNA and Z-RNA"

_molecules, 2023, doi:10.3390/molecules28020843_

Round 1
Reviewer 1 Report
This manuscript of “Formation and stabilization of Z-DNA and Z-RNA” reviews and analyzes the structure, stability, free energy aspects for their formation, especially the comparison between Z-RNA and Z-DNA. It is a nice paper at the time Z-RNA should be paid more attention for both basic research and new targets of medicine. I believe most of the ZNA researches as well as other audience will like this review. The comments are as follows.
1) The title should be modified. For example, it may be changed to “Structure and formation of left-handed duplex of nucleic acids”. Maybe it is better to use a subtitle describing the authors’ claim on Z-RNA.
2) In the Introduction part, an overview about other reviews (especially for the structures) may be added to show the difference and significance of this review.
3) In Figure 1 (and main text related to Figure 1), not only the similarity, the difference (although it is so tiny) in structure between Z-DNA and Z-RNA should be described.
4) It is perfect if the Z-DNA structures involving A-T base pair can be shown (e.g in Figure 2) to relative positions of adjacent base pairs.
5) The syn-conformation of pyrimidines can cause a great position change of hydrogen bonds between base pairs (as compared with syn-purines). This may be the main factors that syn-C or syn-T is unfavorable for non-APP sequences.
6) For any discussion about stability of Z-DNA or Z-RNA, the corresponding factors contributing to the stability of B-DNA or A-RNA should be considered. Formation of ZNA never only comes from the stabilization of ZNA (Please see: Nonalternating purine pyrimidine sequences can form stable left-handed DNA duplex by strong topological constraint. Nucleic Acids Research, 50(2), 684-696). It should be the difference in free energy between ZNA and B-DNA (or A-RNA). The authors did very well for the explanation of stability effect of methylation effect. However, the similar discussion should be added for other factors. For example, binding of cations to B-DNA may be added in Figure 10.
7) For activation energy for transition to ZNA depends greatly on conditions and length of sequence. In some cases (e.g. high salt concentrations), Z-DNA is more stable than B-DNA. Even the difference between Z-DNA and B-DNA is hard to estimate, let alone the activation energy (it changes with situations).
8) About the Junction, more discussion is expected. For example, the topological constraint should be considered. The unfavorable effect to form two junctions should be less than the broken of two base pairs. Especially for RNA cases, because mismatches are also allowable, negative effect of junctions may be less.
9) The color of atoms may be improved in the structures.
10) Lanes 54-56: the “inherent chemical properties” can be changed to “inherent structural properties”, because no chemical reaction of ZNA is discussed.
11) Lane 168: “are shifted away from the center of the molecule” may be changed to “are shifted away from the center of the structure” or “are shifted away from the center of the duplex axis”?
12) Lane 403, 404: In “MgCl2”, “2” should be in shown as a subscript.
13) More references of recent researches on Z-DNA can be added.
14) The style of references should be checked carefully. For example, dot is added for some abbreviations, but not for others. “Nat. Struct. Mol. Biol. 2” and “Proc Natl Acad Sci U S A”.
15) Other careless mistakes should be checked carefully and revised.
Author Response
See uploaded file.

Reviewer 2 Report
This is a timely and comprehensive review on an important emerging field. Krall et al. especially focused on the physico/chemical properties of Z-RNA formation in comparison with the better understood Z-DNA. Z-RNA has been in the limelight recently due to high profile publications in the broader context of innate immunity and clinical relevance in cancer immuno-checkpoint therapies. To have a comprehensive and well written review outlining the current knowledge in chemical factors enabling Z-RNA formation and its energetics is therefore timely and of interest to a broader audience
Author Response
No revision has been request.
Reviewer 3 Report
This is an excellent and thorough review of Z-DNA and, especially, Z-RNA. It provides a remarkably complete presentation of current knowledge along with historical perspectives. I particularly like the section on Hofmeister salts. The review will, hopefully, provide a base and impetus for future research on Z-RNA structure.
I did manage to find one quibble. On line 766 change 'occurs' to 'occuring.'
Author Response
This is an excellent and thorough review of Z-DNA and, especially, Z-RNA. It provides a remarkably complete presentation of current knowledge along with historical perspectives. I particularly like the section on Hofmeister salts. The review will, hopefully, provide a base and impetus for future research on Z-RNA structure.
I did manage to find one quibble. On line 766 change 'occurs' to 'occuring.'
‘Occurs’ has been changed to ‘occurring’ in line 766